# Angular Modeling of the Components of Net Radiation in Agricultural Crops and Its Implications on Energy Balance Closure

Fernando Paz [1], Ma Isabel Marín [1], Jaime Garatuza [2], Christopher Watts [3], Julio Cesar Rodríguez [4], Enrico A. Yepez [2], Antoine Libert [5] and Martín Alejandro Bolaños [6,*]

1 GRENASER, Colegio de Postgraduados, Campus Montecillo, Montecillo 56230, Mexico; ferpazpel@gmail.com (F.P.); isabelmsosa@gmail.com (M.I.M.)
2 Departamento de Ciencias del Agua y Medio Ambiente, Instituto Tecnológico de Sonora, Ciudad Obregón 85000, Mexico; garatuza1@gmail.com (J.G.); yepezglz@gmail.com (E.A.Y.)
3 Departamento de Física, Universidad de Sonora, Hermosillo 83000, Mexico; christopher.watts@correo.fisica.uson.mx
4 Departamento de Agricultura y Ganadería, Universidad de Sonora, Hermosillo 83323, Mexico; jcrod2001@yahoo.com
5 Programa Mexicano del Carbono, Texcoco 56225, Mexico; antoinelibert@hotmail.com
6 Posgrado en Hidrociencias, Colegio de Postgraduados, Campus Montecillo, Montecillo 56230, Mexico
* Correspondence: martinb72@gmail.com or bolanos@colpos.mx; Tel.: +52-59-5952-0200 (ext. 1163)

**Abstract:** Efficient water management in agricultural crops is necessary to increase productivity and adapt to climate change. Evapotranspiration (ET) data are key in determining the water requirements of crops and set efficient irrigation schedules. Estimating ET at the regional scale (for example, in irrigation districts) is a technically complex task that has been tackled by using data acquired by remote sensors on satellites that can be validated with scaled up field measurements when area sources are matched. Energy and matter flux measurements using the eddy covariance (EC) technique are challenging due to balance closure issues, claimed to be due to the different footprints of the energy-balance components. We describe net radiometer footprints in terms of the sun-sensor geometry to characterize the bidirectional distribution functions of albedo and thermal emissions. In this context, we describe a one-parameter model of the components of net radiation that can be calibrated with a single data point. The model was validated in an experiment with five agricultural crops (bean, sorghum, chickpea, safflower, and wheat) at Valle del Yaqui, in Sonora, Mexico, using different sun-sensor geometry configurations. The results from the experimental fits were satisfactory ($R^2 > 0.99$) and support the use of the model for albedo and radiative (surface) temperature in order to estimate net radiation. The analysis of the implications regarding a mismatch among footprints of the components of the energy balance showed that net radiometer fluxes are overestimated most of the time, implying that the closure problem could be solved by using a similar footprint as the aerodynamic components of the energy balance.

**Keywords:** radiative temperature; albedo; footprint; eddy covariance

## 1. Introduction

Surface evapotranspiration (water evaporation from soil and transpiration from plants) is a key process in the exchange of energy and matter between the atmosphere and biosphere. Its contribution to radiative forcing (through water vapor generation and cloud formation) associated with climate change is as important as carbon dioxide emissions from land-use change [1,2]. The geographic distribution and availability of water is the main limiting factor of vegetation growth in about forty percent of the Earth's surface [3]. Water management in irrigated agriculture poses significant challenges for monitoring the water requirements of agricultural crops. Although water requirements in individual

agricultural plots can be reliably estimated using relatively simple techniques [4], the task is not that simple at the regional scale (e.g., irrigation districts).

Remote sensing technologies coupled with energy balance models have been explored as a means for developing methods to estimate ET [5–7]. Several authors have reviewed the different ET estimation schemes using remote sensing [8–11]. They have shown that the major sources of error in estimations include the use of radiative temperature (Tr) in the 8–14 μm interval, instead of the aerodynamic temperature of sensible heat flux (To), the relationship between net radiation (Rn) and soil heat flux (G), and the difficulties in the temporal and spatial scaling of fluxes.

Direct measurements of energy and matter fluxes are commonly carried out using the eddy covariance flux technique [12,13]. With this technique, energy and matter fluxes are measured through the covariance of wind speed in three dimensions and temperature (sensible heat, or H) or water vapor (latent heat, or λE, where λ is the heat of vaporization of air). The area of influence or footprint of such measurements is dynamic and varies with sensor height, wind speed and direction, morpho-structural features of vegetation, and atmospheric stability [14–16]. On the other hand, Rn has a constant footprint that depends on the viewing angle and observation height of the sensor [17]; G has a fixed geometric configuration of sensors (heat plates) on the ground, so that it also has a constant footprint. The different measurement footprints of the components of energy balance may account for the commonly observed issue of lack of closure, which ranges between 10 and 30% [18,19]; estimated typical error between 5 and 50% of the components of the energy balance equation. This situation, in addition to the spatially-defined footprint of remote sensing products, challenges scaling the EC flux measurements to make them comparable with data acquired by remote sensors [20,21]. Different approaches have been used to scale (by aggregation/disaggregation) flux measurements [22,23] with acceptable results, but they are difficult to replicate in practice due to the complexity of the parameterization (i.e., knowledge of geometry of crops and parameters for the partition of fluxes), which is not available when using remote sensing techniques at large scales.

From the perspective of energy balance closure using EC, it is necessary to match footprints of different components in order to have a correct evaluation of the energy closure problem [17]. For this reason, a valuable contribution to solve this problem is to have a simple model of the net radiation footprint under oblique views without knowledge of the geometry of vegetation or the partition of fluxes associated with this geometry. Although there are several models of footprint modeling of components of net radiation [24–27], for angular (directional) estimations, all of them require specific data of the geometry of crops that it is not available in remote sensing approaches. From this perspective, a hypothesis to be tested is that a net radiometer footprint can be modeled using a one-parameter model (plus angular data), thus requiring one measurement to be parameterized. This paper discusses the issue of energy balance closure resulting from differences in the footprint components. We propose a model to characterize Rn in terms of the basic components: albedo and surface temperature/emissivity under a sun-sensor geometry. We discuss the development of the footprints of these components, based on a simple model parameterized under sun-sensor geometry considerations. As the worst balance energy closures are observed in crops [28], the model proposed was validated in a field experiment with five different agricultural crops (bean, sorghum, chickpea, safflower, and wheat) at Valle del Yaqui in Sonora, Mexico. The implications of the developed model are discussed for the energy balance closure problem, particularly on the consequences of using a fixed footprint for net radiation.

## 2. Materials and Methods

### 2.1. Energy Balance and Net Radiation

The balance of energy fluxes (all components expressed in W m$^{-2}$) on a surface is given by:

$$Rn = \lambda ET + H + G \tag{1}$$

Under certain conditions, λET can be estimated directly from Rn using the Priestley and Taylor (1972) [29] coefficient [30]. The balance energy closure (CEB ≤ 1, generally) is given by [28]:

$$C_{EB} = \frac{\lambda E + H}{Rn - G} \tag{2}$$

A value of 1.0 denotes a perfect closure. The closure of energy balances is part of the quality control of eddy covariance measurements [31] and has been reviewed in several publications [18,19,28,32]. From the analysis of the closure problem in EC measurements [18], this can be related to an overestimation of available energy (Rn–G) or an underestimation of aerodynamic fluxes (λET + H). The energy balance closure can be related to several causes [18,32]: (a) errors in sampling with different sources of areas of sensors, (b) systematic errors of the instrumentation used, (c) energy sources (storage components) not considered, (d) losses of contributions due to low or high frequencies, and (e) advection of scalars.

In flux measurements with eddy covariance, there are different footprints of the energy balance components for a given wind direction, with Rn, H, and λET sensors in a fixed position, across a homogeneous agricultural plot. Using footprint measurements to make inter-comparisons of energy balance components is only valid for uniform, dense vegetation. Arguably, uniform dense vegetation is not the norm in natural ecosystems, but may be achieved in agricultural crops. Figure 1 shows the geometric configuration of plants in a ridge-and-furrow agricultural plot, with alternating zones of bare soil and total coverage by the crop. The difference in the geometric arrangement of plants and the geometry of observations (footprints) creates heterogeneity (the footprints observe/measure different portions of vegetation and soil) in an otherwise "homogeneous" plot; this leads to important differences in measurements recorded previous to a uniform and dense condition over the entire plot.

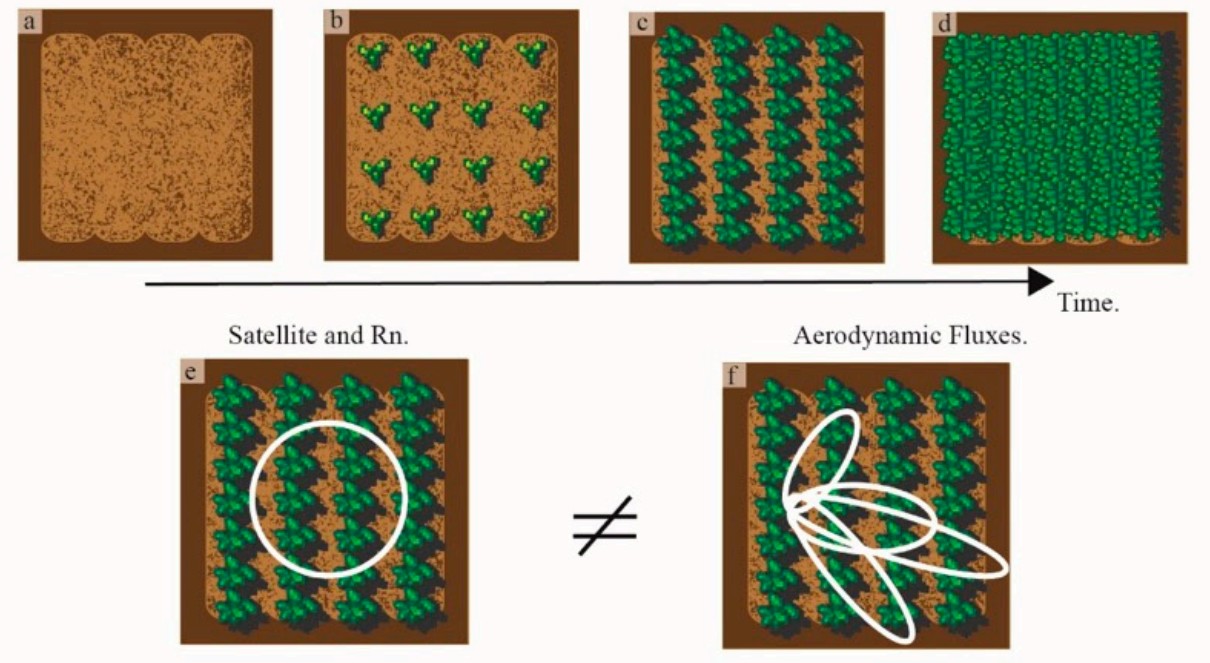

**Figure 1.** Geometric configuration of plants in a ridge-furrow agricultural crop across time (**a–d**) and the complex differences between footprint from different observations; (**e**) satellite and in situ Rn measurements; and (**f**) flux-based methods such as eddy covariance.

The ground heat flux (G) component is measured using heat plates placed on the ground in a fixed geometric arrangement (see Figure 2), which results in a constant footprint. The sun-lit and shaded parts of soil and vegetation vary throughout the day and over the

growth cycle in relation to crop growth and sun-sensor geometry. Thus, it is inadequate to assign equal weights to the different heat plate sensors when estimating G.

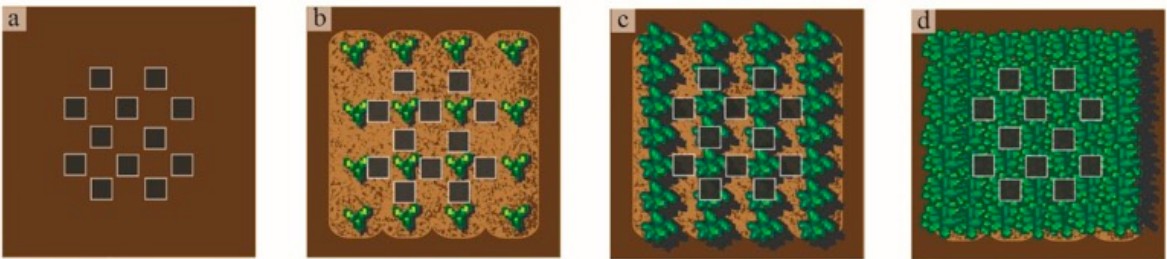

**Figure 2.** Arrangement of heat plate sensors on the ground for different growth stages (time progress) of a crop.

To estimate G, under a given geometric configuration of heat plate sensors, modeling plant growth in a field crop is relatively straightforward based on their shape, geometric arrangement in the plot, and sun-sensor geometry. Thus, the value of G corresponding to any footprint with a given orientation and dimension is obtained as:

$$G_{\text{footprint}} = \frac{(\text{PMvi})(\text{Avi}) + (\text{PMvs})(\text{Avs}) + (\text{PMsi})(\text{Asi}) + (\text{PMss})(\text{Ass})}{\text{Avi} + \text{Avs} + \text{Asi} + \text{Ass}} \tag{3}$$

where PM is the average measurements over the sun-lit or shaded conditions and As is the area of sun-lit or shaded conditions in the footprint, as indicated by subindices vi = sun-lit vegetation, vs. = shaded vegetation, si = sun-lit soil, and ss = shaded soil.

The components of net radiation are given by:

$$\text{Rn} = (\text{Rs} \downarrow - \text{Rs} \uparrow) + (\text{Rt} \downarrow - \text{Rt} \uparrow) \tag{4}$$

where Rs is the short-wave solar radiation (W m$^{-2}$), Rt is the long-wave (thermal) radiation (W m$^{-2}$). Arrows denote whether radiation is incoming (downward) or outgoing (upward).

Equation (4) can be reformulated using the Stefan–Boltzmann equation) [33], as follows (removing the arrows):

$$\text{Rn} = (1 - \alpha)\text{Rs} + (\varepsilon_a \sigma T_a^4 - \varepsilon_s \sigma T_s^4) \tag{5}$$

where $\alpha$ is the surface albedo (dimensionless); $\varepsilon s$ is the surface emissivity (dimensionless); $\sigma$ is the Stefan–Boltzmann constant (= 5.67 × 10−8 W m$^{-2}$ K$^{-4}$); $\varepsilon_a$ is the air emissivity (dimensionless); Ts (K) is the surface temperature; and Ta (K) is the air temperature. Ts equals the surface radiative temperature.

Air emissivity can be estimated from [34]:

$$\varepsilon_a = 1.24 \left( \frac{e_a}{\text{Ta}} \right)^{1/7} \tag{6}$$

where $e_a$ is the vapor pressure of air (hPa).

Sun radiation data can either be obtained from weather stations or estimated using remote sensing [35].

Satellite-borne remote sensors measure the radiance (L) in thermal wavelength bands, from which temperature can be calculated using Planck's equation:

$$\text{Lcn} = \frac{C_1}{w^5 \pi \left[ \exp\left( \frac{C_2}{w\text{Tb}} \right) - 1 \right]} \tag{7}$$

where Lcn (W m$^{-2}$ µm$^{-1}$) is the black-body radiance ($\varepsilon = 1$); w is the wavelength (µm); Tb (K) is brightness temperature; $C_1$ = 3.74151 x 10$^{-16}$ (W m$^{-2}$); and $C_2$ = 0.0143879 (m K).

Surface emissivity is calculated as the ratio between surface radiance (Ls) and black-body radiance (Lcn):

$$\varepsilon_s = \frac{Ls}{Lcn} \qquad (8)$$

To simplify the description, we omitted the spectral ($\lambda$) and angular arguments. Accordingly, it was assumed that T, L, and $\varepsilon$ are measured in a bandwidth similar to the longwave spectral region, or that this can be estimated using small bandwidths. For albedo, the bandwidth corresponds to the shortwave segment of the electromagnetic spectrum (0.25–3 µm).

Field measurements show that surface Tr varies with sun-sensor geometry: $\Psi = (\theta v, \phi v, \theta s, \phi s)$, where $\theta$ denotes zenith angles and $\phi$ azimuth angles; v denotes viewing and s denotes sun illumination [36,37]. Similarly, surface emissivity has angular effects similar to Tr (or Ls) [38,39]. Considering both variables (Tr and $\varepsilon$), the bi-directional thermal emission distribution function (BEDF) [40] should be known in order to model the angular effects of the sun-sensor geometry. Although different modeling schemes are available [27,41–43], all are difficult to parameterize, particularly with a single data point (one measurement). Angular variations in BEDF can be used to estimate sensible heat using two-source models [44], but this approach has not been extended to net radiation components.

Field measurements show significant angular effects on surface albedo [45]. Therefore, it is necessary to model the bi-directional distribution function of reflectance (BRDF) or albedo. BRDF models [26,46], are also difficult to parameterize.

Finally, the relationship between the footprints of field measurements (Rn and G) associated with sun-sensor geometry should be determined. Figure 3 shows how the area (ellipses for oblique zenith angles and circles for a nadir view) of a sensor measurement changes according to the viewing zenith angle (the direction of the ellipse's major axis is a function of the viewing azimuth angle) for a given sun illumination condition. Paz and Marin (2019) [47] show how to calculate geometry using the variables shown in Figure 3.

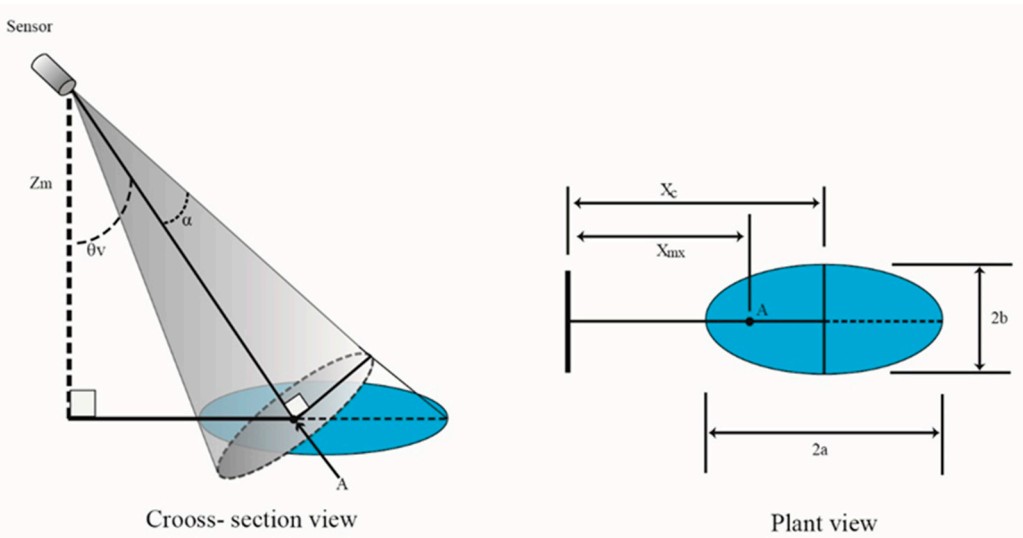

**Figure 3.** Footprint associated with the viewing geometry of the sensor where the footprint varies with view zenith angles [47].

Considering that aerodynamic footprints are larger than Rn and G [48], and therefore a mismatch problem [17,18], it is necessary to have similar footprints of all components of the energy balance. In crops and grasslands, [48] estimated that it is necessary to have a Rn sensor at a height between 6–15 times higher than the aerodynamic fluxes to have similar footprint areas, which create new logistic and interpretation issues and yet, do not consider angular variations in source areas.

The Rn footprint under nadir view is circular (Figure 1a) but for oblique views (changing view zenith angles), it is elliptical [17,47]; extended the circular footprint for this condition.

*2.2. One-Parameter Model for BRDF and BEDF*

In this section, we discuss how normalized spaces can be used to reduce the complexity in measuring the energy balance of the sun-sensory geometry components as well as to estimate BRDF and BEDF, thus modifying the energy and matter balance equation.

The one-parameter model (OPM hereafter) for BRDF and BEDF is a modeling scheme that differs from other models currently used in operational applications of remote sensing. The OPM considers a particular symmetry (a hot spot, at the point where the viewing zenith and illumination angles coincide), which simplifies the modeling of BRDF and BEDF into a single parameter, so that only one data point (a single field measurement plus angular data) is required and applicable at the pixel level for any satellite image acquired any time. The OPM was initially developed for modeling reflectance using particular symmetries for the different spectral bands [49]; it was afterward generalized into a single symmetry for all bands [50]:

$$
\begin{aligned}
\chi &= 90 - \theta v + \theta s \\
Vn &= f(V)\cos(\chi) \\
\chi &= a - gRn
\end{aligned}
\tag{9}
$$

where V can be albedo, emissivity, or reflectance in any band in the shortwave electromagnetic spectrum, or radiative temperature on any thermal band; g is a parameter corresponding to BEDF or BRDF; and $a = 90°$. The function f(V) equals ln (V) when the scale effect is taken into account (that is, the area changes with the viewing zenith angle); f(V) equals V when the scale effect is not considered.

Under the same assumptions, the BRDF or BEDF model defined by Equation (9) can be extended to the case of azimuth angles:

$$
\begin{aligned}
d\phi &= \phi v - \phi s \\
\text{If } d\phi &\leq 180, \ d\phi p = d\phi \\
\text{If } d\phi &> 180, \ d\phi p = 360 - d\phi \\
\text{If } d\phi p &\leq 90, \ \xi = d\phi p + \theta s \\
\text{If } d\phi p &\leq 90, \ \xi = d\phi p + \theta s \\
gn &= g\cos(\xi) \\
\xi &= A - G(gn)
\end{aligned}
\tag{10}
$$

where G is the parameter corresponding to BRDF or BEDF and $A = 90°$.

The OPM allows for parameterizing BRDF and BEDF with a single parameter: g for cases where only the zenith angle (viewing nadir angle) varies, or G for the general case. Constants a and A equal $90°$ as a result of the symmetry implied by the position variables $\chi$ and $\xi$.

The system of Equations (9) and (10) can be reformulated ($a = 90$, $A = 90$) as:

$$
f(V) = G\left(\frac{90 - \chi}{90 - \varsigma}\right)\left[\left(\frac{\cos(\xi)}{\cos(\chi)}\right)\right]
\tag{11}
$$

The OPM has been validated for reflectance measurements in experimental settings in the laboratory [51,52] and with variations in the angular vision of satellite reflectance [53,54], while also being used to estimate more complex BRDF models [55] with good results throughout (generally, $R^2 > 0.99$).

Statistics commonly used for the empirical assessment of the fit of the OPM to field data include the root mean square error (RMSE) and the mean absolute error (MAE) and mean relative error (MRE):

$$\text{RMSE} = \left\{ \frac{1}{n} \sum_{i=1}^{n} (Tn, med - Tn, est)^2 \right\}^{0.5} \tag{12}$$

$$\text{MAE} = \left\{ \frac{1}{n} \sum_{i=1}^{n} \left| \frac{Tn, med - Tn, est}{Tn, med} \right| \right\} \times 100 \tag{13}$$

$$\text{MRE} = \left\{ \frac{1}{n} \sum_{i=1}^{n} \frac{Tn, med - Tn, est}{Tn, med} \right\} \times 100 \tag{14}$$

Additionally, a simple linear regression was fit to the measured (med) and estimated (est, using the OPM) Rn values:

$$Rn, est = c + dRn, \ med \tag{15}$$

Parameters c and d as well as the statistic $R^2$ were calculated for all the dates when albedo and radiative temperature were measured under different sun-sensor geometries.

### 2.3. Study Area

The study was carried out in 2008 in an agricultural field located in Irrigation District 041-Río Yaqui, in Sonora, Mexico. The study area is located between coordinates $27°14'24'' - 27°16'48''$ N and $109°52'12'' - 109°54'36''$ W. The data obtained in the experiment were used for various studies including the estimation of above-ground biomass and yield of crops [56], modeling stress in crops [57], biophysical and spectral scaling [58], and modeling energy balances using satellite data [59].

Five homogeneous plots (PH) were chosen to characterize the footprint of $\alpha$ and *Tr*. Plots were considered homogeneous, since a single crop had been planted throughout the plot, on the same date, with a uniform planting density, and under the same furrow spacing and directions. The initial conditions of the selected plots are described in Table 1.

**Table 1.** Initial conditions of the study plots.

| Plot | Surface Area (Ha) | Crop | Furrow Orientation | Furrow Spacing (cm) | Plant Height at the Start of the Experiment (cm) |
|------|------|------|------|------|------|
| PH1 | 89.9 | Bean | North–South | 160 | 0 |
| PH3 | 38.75 | Sorghum | East–West | 80 | 0 |
| PH4 | 38.86 | Chickpea | North–South | 80 | 30 |
| PH5 | 9.59 | Safflower | North–South | 80 | 6 |
| PH6 | 47.97 | Wheat | North–South | 100 | 70 |

### 2.4. Instrumentation

Two measurement schemes were used in each plot. The first scheme aimed to characterize the footprint of $\alpha$ and *Tr* by simultaneously measuring crop reflectance and radiative temperature using different sun-sensor geometry configurations. The measurements were made using an ad hoc system consisting of the following:

(a) A metallic stand for accurately positioning the height of each sensor. The stand consists of an extensible mast with clamping mechanisms at 2.5 m, 4.0 m, and 5.5 m.

(b) A polycarbonate structure with a flat base to support the Tr sensor. The structure was fitted with an electronic mechanism with servo motors to accurately position the Tr sensor at any angle between 0° and 90° in the zenith plane (Figure 4a) and between 0° and 180° in the azimuth plane (Figure 4b).

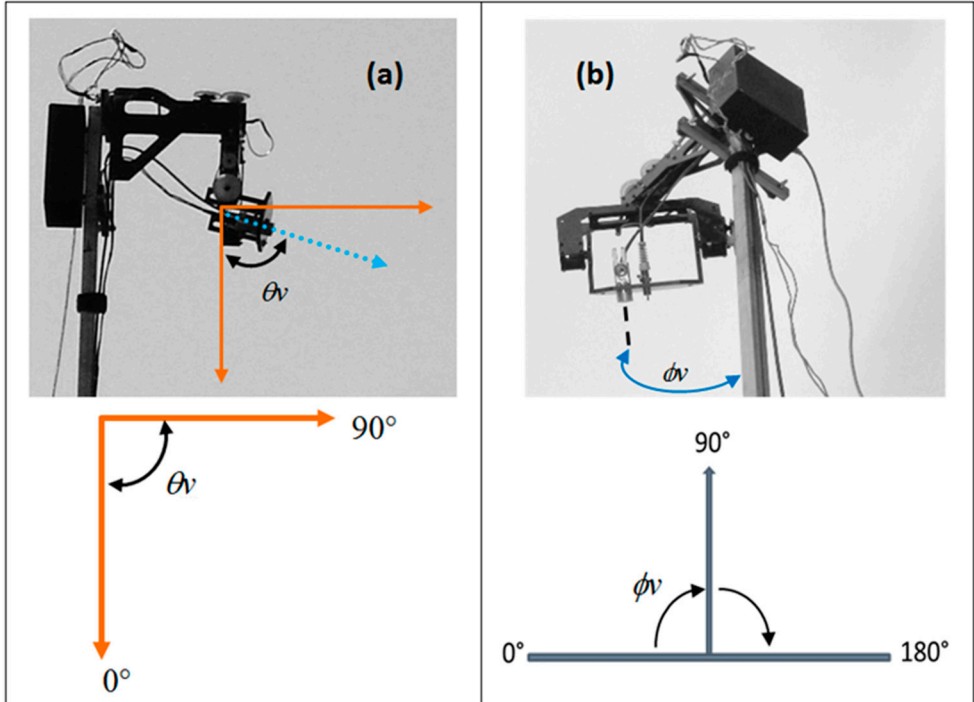

**Figure 4.** Geometry of the footprint-measuring device. (**a**) Zenith plane, (**b**) azimuth plane.

(c)    Control card and software for operating the polycarbonate structure.
(d)    Crop reflectance sensor. A hyperspectral (continuous data in 2 nm-wide bands on the 350 to 2500 nm region) radiometer with a 25° viewing angle (ASDTM; FieldSpecFR Jr optical fiber).
(e)    Radiative temperature sensor for the crop. Apogee™ model IRTS infrared thermometer with an 18.4° viewing angle (3:1 sensor height viewing:diameter ratio).
(f)    Console for system operation and data storage.

The measurement device was mounted on a tripod modified with a central support to stabilize the measuring system. A bubble level was used to maintain the tripod was level; this ensured a vertical position during measurements (the equipment was disassembled at the end of each day and reassembled on the following day because of measurement rotation among crops). The measurement system was mounted with a 90° azimuth angle, parallel to the furrow direction. Efforts were made for the measurement site to represent the conditions prevailing in the crop, avoiding any disturbance features. However, for practical reasons (transport, equipment assembly and disassembly), it had to be located toward one end (but several meters inside) of the plot, leaving a margin wide enough to prevent the measurements from being affected by the adjacent bare soil surface. A mark was left on each plot to easily locate the same site on subsequent weekly visits. The initial conditions of the study plots are described in Table 1.

### 2.5. Experimental Design

Each experimental plot had a EC system, but with only one heat plate or none for the measurement of G. Due to this lack of information, no intent was conducted for energy balance closure estimation. A net radiometer (CNR1, Campbell Scientific) was placed at a 3.0 m height connected to a datalogger (CR500, Campbell Scientific) for the storage of measurements sampled at 10 Hz. All data were averaged to half hour intervals.

The experimental campaign was carried out during the whole crop cycles between February and May 2008. Each plot was visited once a week, and three measurement cycles of crop reflectance and radiative temperature were carried out on each visit. Measurements were made using the heights and viewing angles ($\phi v, \theta v$) listed in Table 2.

**Table 2.** Geometric parameters used for the measurement of radiative temperature and albedo.

| Sensor Height (m) | Azimuth Angle ($\phi v$) | Zenith Angle ($\theta v$) | No. of Readings |
|---|---|---|---|
| 2.5 | 15°, 45°, 90°, 135° y 165° | 40°, 60°, 70°, 75° | 20 |
| 4.0 | 15°, 45°, 90°, 135° y 165° | 20°, 40°, 60°, 70°, 75° | 25 |
| 5.0 | 15°, 45°, 90°, 135° y 165° | 20°, 40°, 60°, 70°, 75° | 25 |

As shown in Table 2, for a given sensor height and azimuth position, reflectance and radiative temperature readings were obtained by varying the zenith position of the sensor. Thus, for example, with the sensor at 2.5 m height and 15° azimuth angle, four reflectance and Tr readings were made at zenith inclinations of 40°, 60°, 70°, and 75°. In total, 70 readings were made in each measurement cycle.

It should be pointed out that the azimuth plane of our system (Figure 4b) differed from the common practice (azimuth angles are positive in a clockwise direction from North). Azimuth used in the experimental measurements used a value of zero perpendicular to rows of each crop field. The correspondence between the azimuth of the system and the actual azimuth depends on the location of the measurement site on each PH. All azimuths used in the field were changed to the standard notation and this convention was used in this paper.

The measurement cycles were carried out at different times of the day to obtain data for different solar angles (zenith and azimuth). The first cycle was carried out in the morning, the second around solar noon, and the third near sunset. Each measurement cycle was scheduled in advance so that they provide measurements: (1) representing three different sun zenith angles; and (2) with a difference of at least 10° between the selected zenith angles.

### 2.6. Measurement of Footprints

Based on instrument positioning and geometrical angles, differences in the expected source areas are illustrated in Figure 5, which depicts the footprints of the reflectance and radiative temperature measurements in relation to sensor height and viewing zenith angle. The parameters of ellipses that represent footprints are summarized in Table 3.

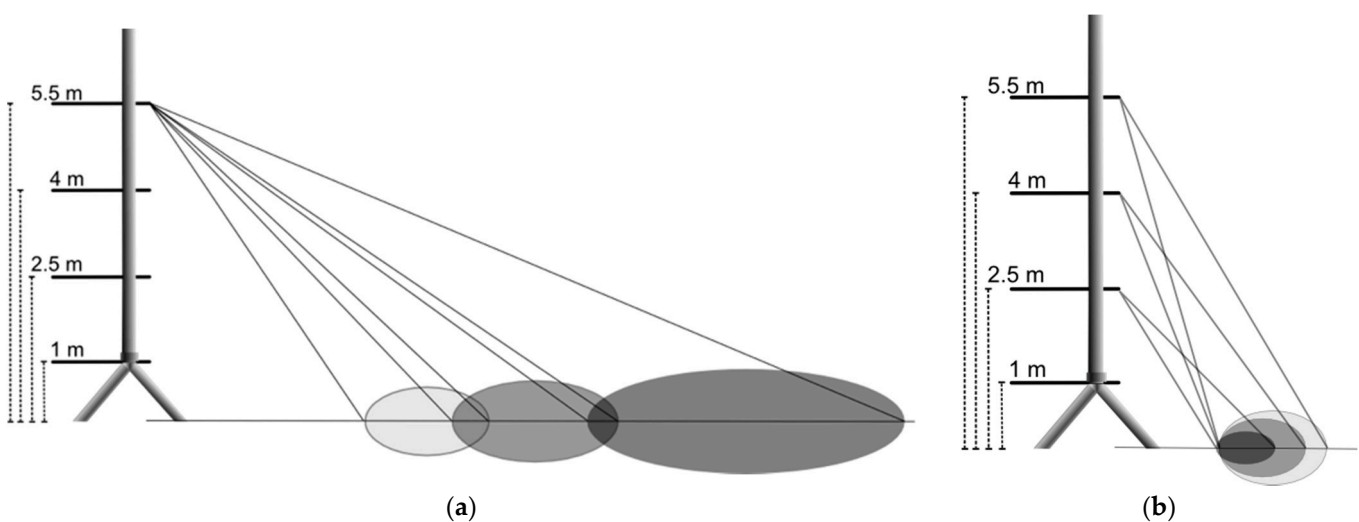

**Figure 5.** Footprints of reflectance and radiative temperature measurements for (**a**) a fixed height and varying viewing zenith angles, or (**b**) a fixed viewing zenith angle and different heights.

**Table 3.** Areas of influence of the Apogee$^{TM}$ sensor for an 18.4° viewing angle and ASD$^{TM}$ sensor for a 25° viewing angle with three different heights (parameters are described in Figure 3).

| Sensor | Height (m) | $\theta v$ (°) | 2a (m) | 2b (m) | Area (m$^2$) |
|---|---|---|---|---|---|
| | 2.5 | 40 | 1.45 | 1.11 | 1.26 |
| | 2.5 | 60 | 3.63 | 1.82 | 5.18 |
| | 2.5 | 70 | 9.00 | 3.09 | 21.80 |
| | 2.5 | 75 | 20.23 | 5.26 | 83.64 |
| | 4.0 | 20 | 1.51 | 1.42 | 1.69 |
| | 4.0 | 40 | 2.31 | 1.77 | 3.22 |
| Apogee | 4.0 | 60 | 5.81 | 2.91 | 13.27 |
| | 4.0 | 70 | 14.39 | 4.94 | 55.81 |
| | 4.0 | 75 | 32.37 | 8.42 | 214.11 |
| | 5.5 | 20 | 2.08 | 1.96 | 3.20 |
| | 5.5 | 40 | 3.18 | 2.44 | 6.09 |
| | 5.5 | 60 | 7.99 | 4.00 | 25.08 |
| | 5.5 | 70 | 19.79 | 6.79 | 105.52 |
| | 5.5 | 75 | 44.51 | 11.58 | 404.81 |
| | 2.5 | 40 | 1.96 | 1.50 | 2.31 |
| | 2.5 | 60 | 5.20 | 2.61 | 10.66 |
| | 2.5 | 70 | 15.07 | 5.20 | 61.52 |
| | 2.5 | 75 | 52.46 | 13.80 | 568.79 |
| | 4.0 | 20 | 2.02 | 1.90 | 3.02 |
| | 4.0 | 40 | 3.13 | 2.40 | 5.90 |
| ASD | 4.0 | 60 | 8.32 | 4.18 | 27.29 |
| | 4.0 | 70 | 24.11 | 8.32 | 157.50 |
| | 4.0 | 75 | 83.94 | 22.09 | 1456.10 |
| | 5.5 | 20 | 2.78 | 2.61 | 5.70 |
| | 5.5 | 40 | 4.30 | 3.30 | 11.16 |
| | 5.5 | 60 | 11.44 | 5.74 | 51.60 |
| | 5.5 | 70 | 33.14 | 11.44 | 297.77 |
| | 5.5 | 75 | 115.42 | 30.37 | 2752.94 |

2a = Major axis of the ellipse, 2b = Minor axis of the ellipse.

The parameters of ellipses representing the areas of influence of radiative temperature and reflectance are shown in Table 3, calculated according to [47]. All measurements were made within homogeneous plots.

Vegetation albedo was estimated using the relationship between albedo and the spectral bands (B) of the ETM + sensor on board Landsat 7 [60,61] (as proportions of reflectances):

$$\alpha = 0.356B1 + 0.130B3 + 0.373B4 + 0.085B5 + 0.072B7 - 0.0018 \qquad (16)$$

From the spectral measurements for each band of the ETM + sensor, reflectance values were estimated using the spectral response functions corresponding to this sensor. The response function provided by the manufacturer was used to estimate *Tr*.

A failure in the optical fiber of the ASD$^{TM}$ sensor occurred several weeks after the measurements started. This failure interrupted the reflectance measurements for wavelengths below 1000 nm. For this reason, we used the complete dataset ($n$ = 2912) of the overall measurements to fit a multiple regression to estimate the relationship between albedo and bands B5 and B7 of the ETM + sensor, as these were the only bands that could still be recorded after the optical fiber failure.

To estimate the albedo, a relationship was obtained using complete reflectance measurements and bands 5 and 7 of the ETM + sensor ($R^2$ = 0.954) in %:

$$\alpha = 2.754 + 1.754B5 - 1.503B7 - 0.0140B5xB7 + 0.0202B7^2 \qquad (17)$$

In order to analyze the albedo estimations under the optical fiber problem, Figure 6 compares the results obtained with the two models (16 and 17). The model fitted shows good agreement regarding the use of the complete model with minimum bias.

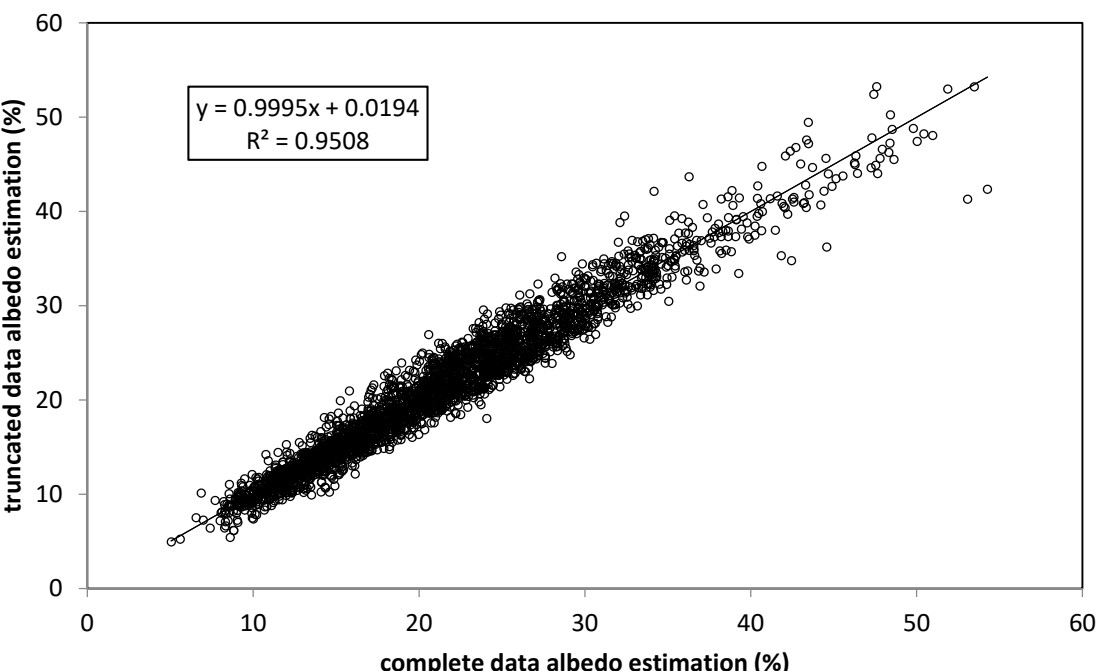

**Figure 6.** Relationship between albedo values estimated using either all of the spectral bands of the ETM + sensor (complete model) or bands B5 and B7 only (truncated model).

## 3. Results

### 3.1. Model Adjustments

To examine how Tr and α vary in relation to sun-sensor geometry, Figure 7 shows the measurements of above-ground coverage (fv) [62] made in PH4 (chickpea) between Julian days 59 and 129, with a coverage peak on day 80.

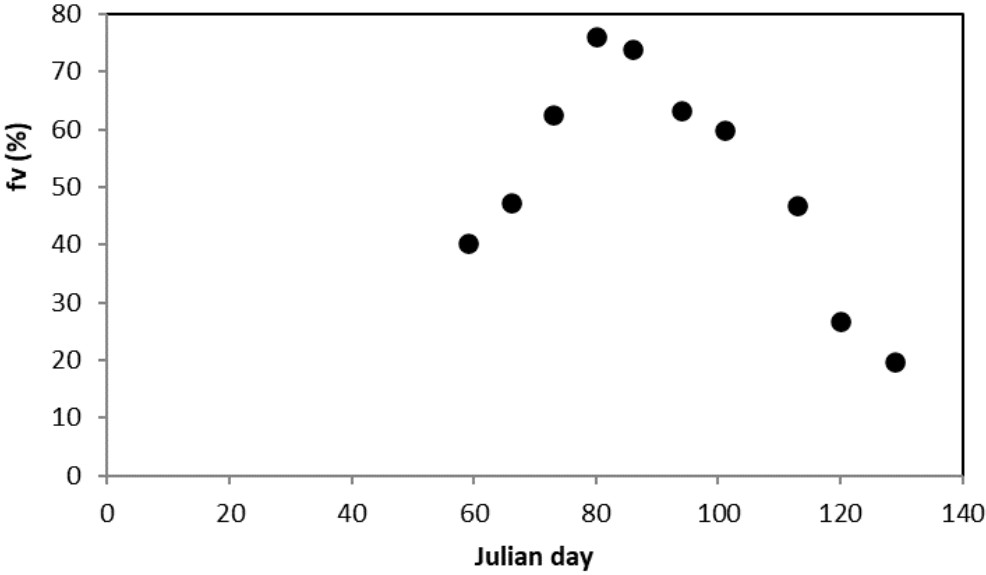

**Figure 7.** Temporal variations in above-ground coverage in PH4 (chickpea).

We analyzed the use of the three different lighting conditions (morning, noon, and afternoon) used in the experiments with various viewing azimuth and zenith angles (Table 2). Variations in Tr and α at the beginning and end of the measurements and during the fv peak are shown in Figure 8 for an observation height of 5.5 m.

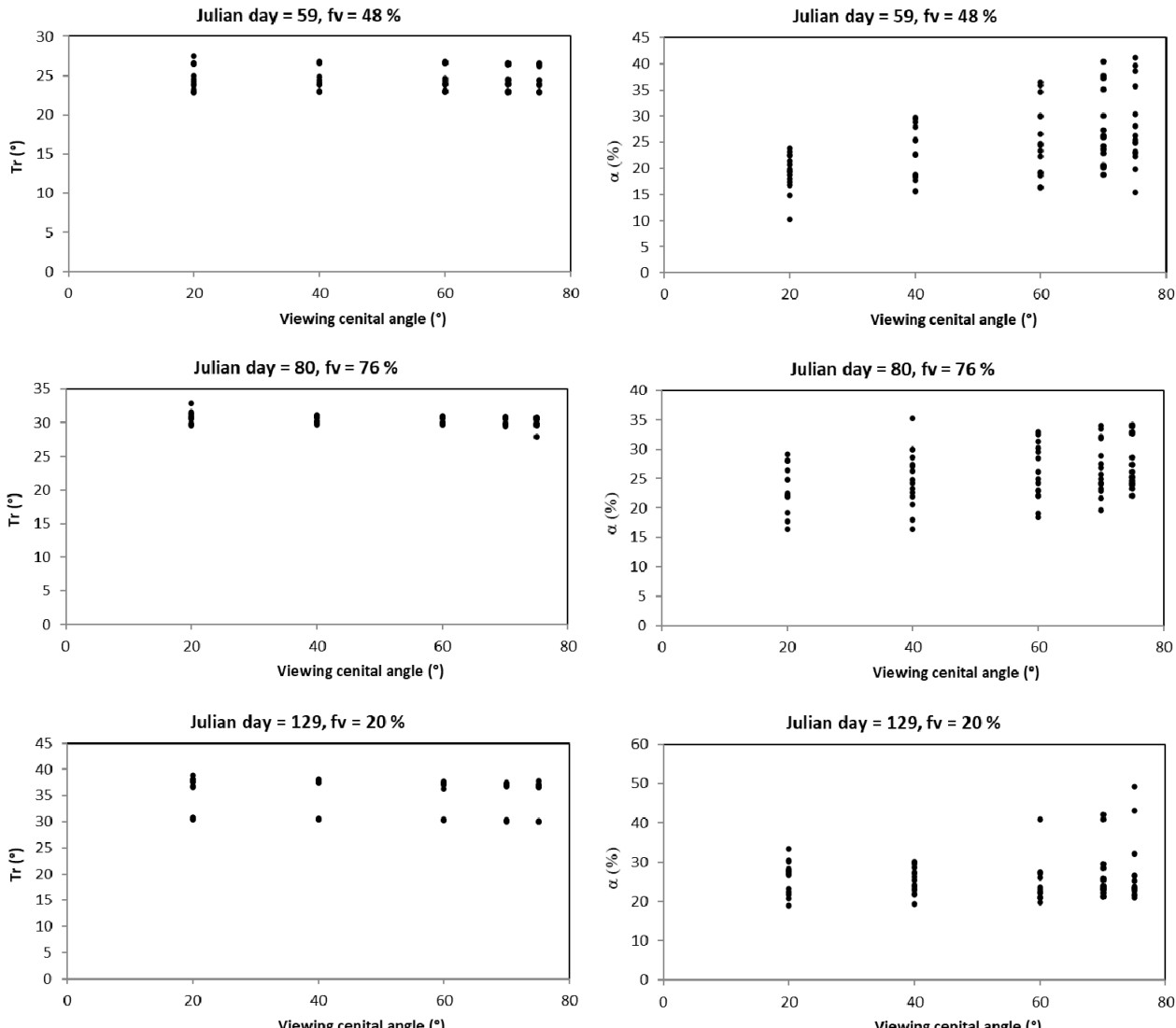

**Figure 8.** Variation of radiative temperature and albedo measurements in PH4 (chickpea) on three different days during the crop growth cycle.

The variations in radiative temperature were up to 10 °C and between 10 and 40% in the albedo values (Figure 8) at the same crop environmental conditions, but different footprint (because of different sun-sensor geometry). This highlights the errors that can occur if measurements are not standardized to a common sun-sensor geometry (same footprint), which, in turn, may lead to large errors in energy flux estimates if the Rn (and G) footprint is different from the aerodynamic fluxes.

The model of sun-sensor geometry was adjusted to minimize estimation error. Figure 9 shows the estimates of radiative temperature and albedo, normalized (*n*) as per Equation (9), for the case with no scale effect; Figure 10 shows the same for the case involving a scale effect.

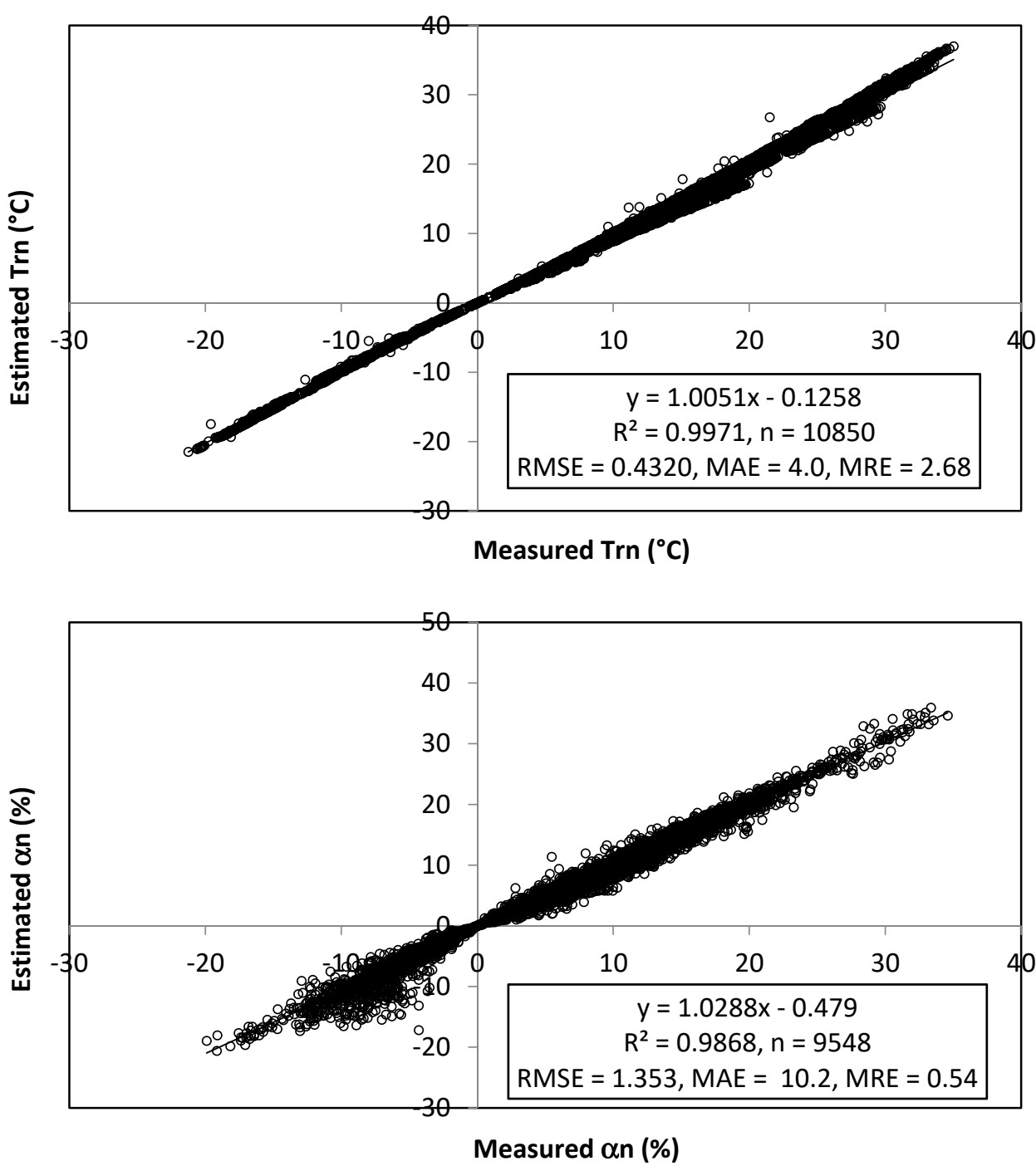

**Figure 9.** Normalized estimates of radiative temperature and albedo for measurements in all PHs during the measurement campaign with no scale effect.

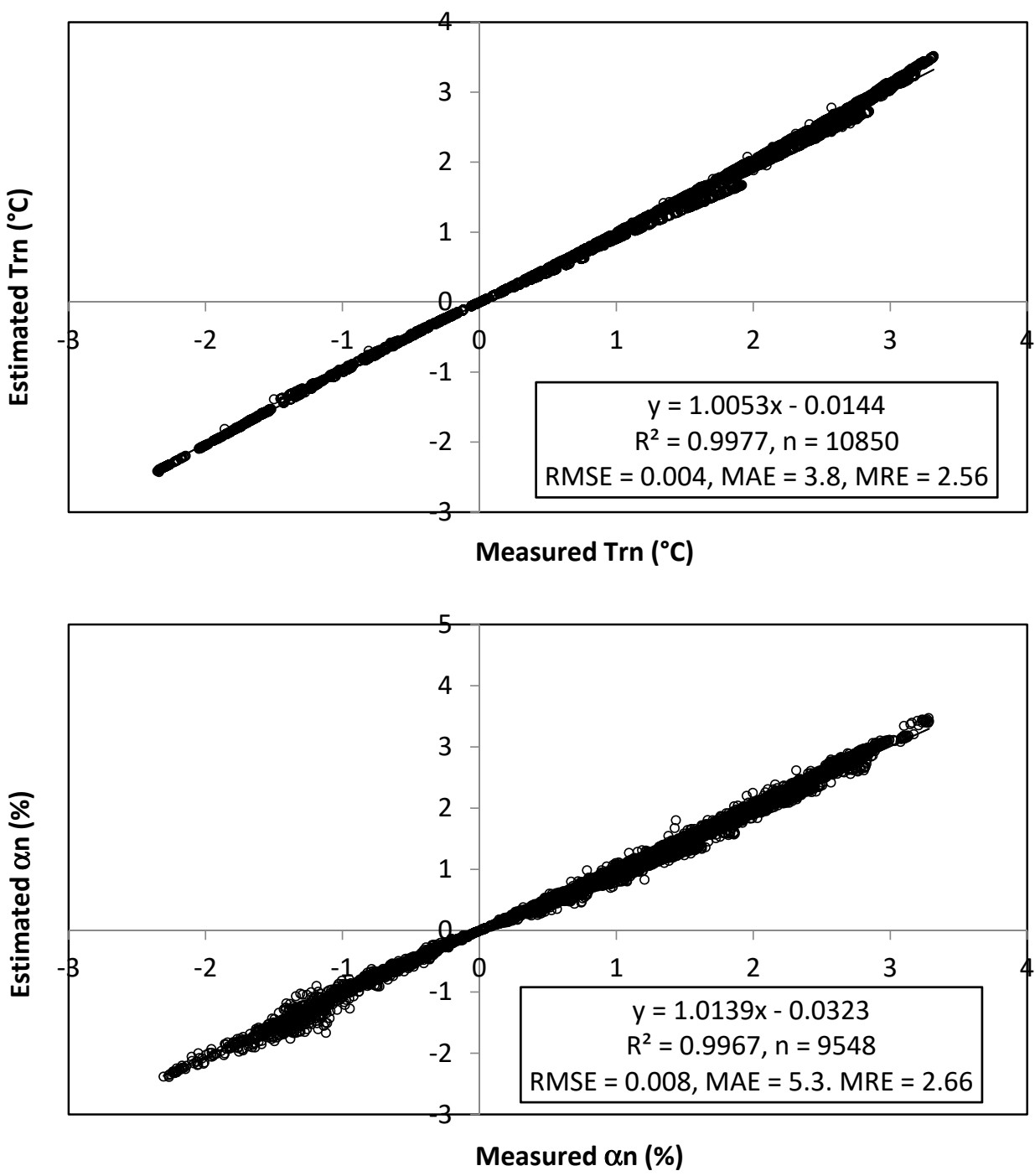

**Figure 10.** Normalized estimates of radiative temperature and albedo for measurements in all PHs during the measurement campaign including the scale effect.

### 3.2. Patterns of Variation with View and Sensor Zenith Angles

To explore albedo and radiative temperature measurement variations with footprint area (viewing zenith angle) under fixed solar illumination, Figure 11 shows the patterns (solar zenith angle = 59.48°) for chickpea (PH4) in a day (only positive Rn measurements), depending on sun-sensor geometry (and crop grow), although the patterns shown in Figure 11 can change depending on fv. Nevertheless, the patterns shown are representative of conditions before maximum fv of the crop (minimum variations).

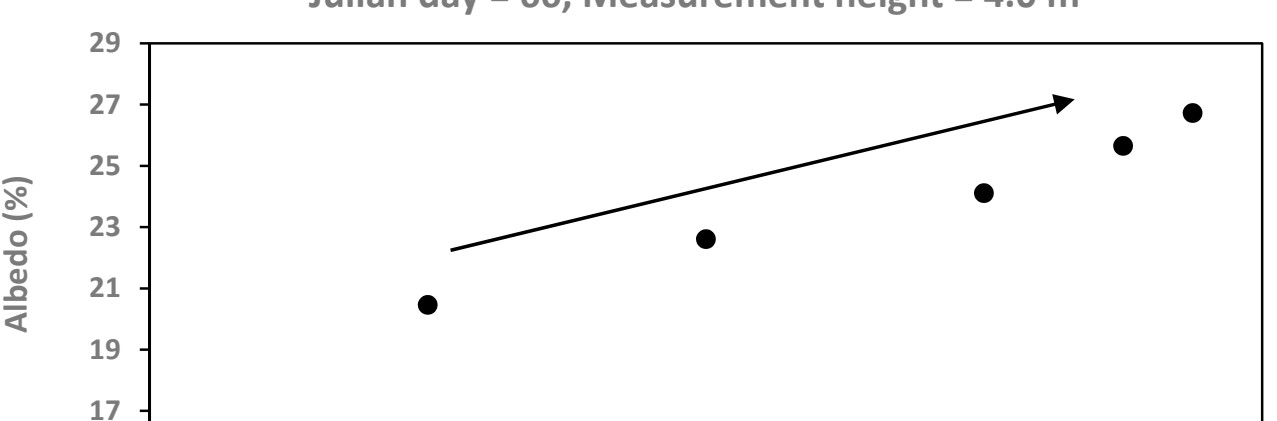

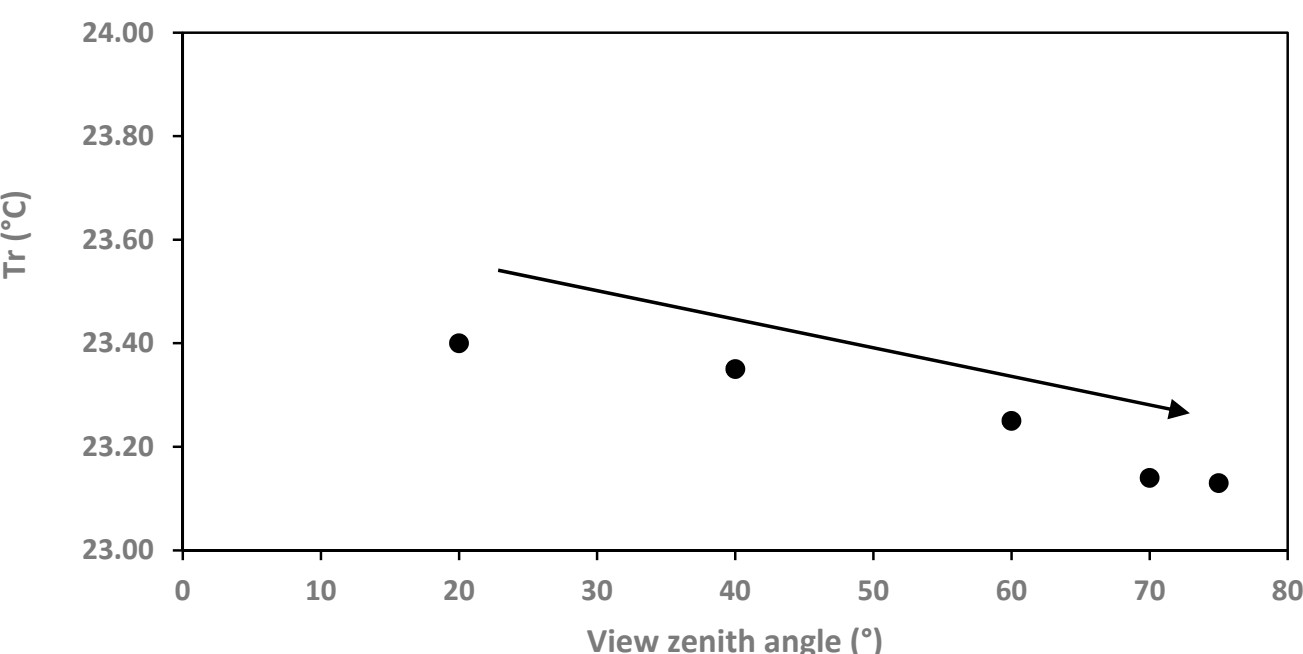

**Figure 11.** Variation of albedo and radiative measurements for chickpea (PH4) for Julian day 66.

Finally, two conditions (fv = 0 and maximum fv) were considered for the analysis of Rn patterns with solar illumination (solar zenith angle variation). The patterns are shown in Figure 12 for chickpea crop (PH4). The maximum value of Rn was around solar moon and it decreased in the morning and afternoon (solar zenith angles were higher than the solar moon).

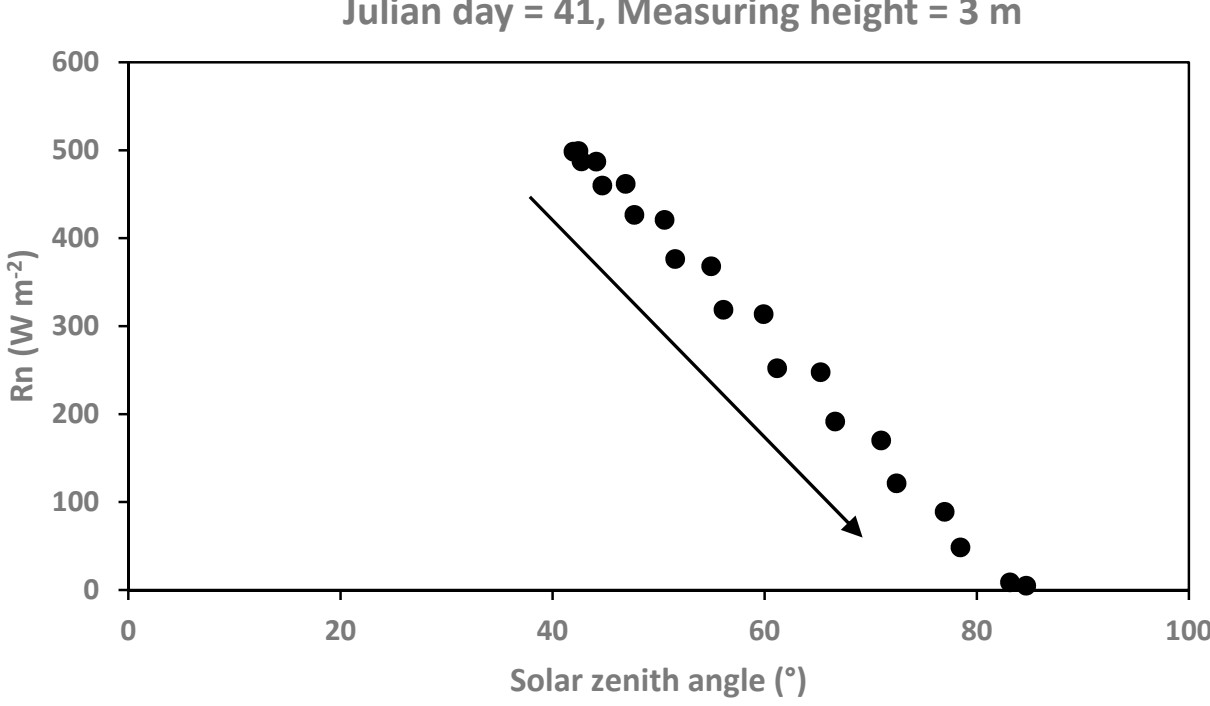

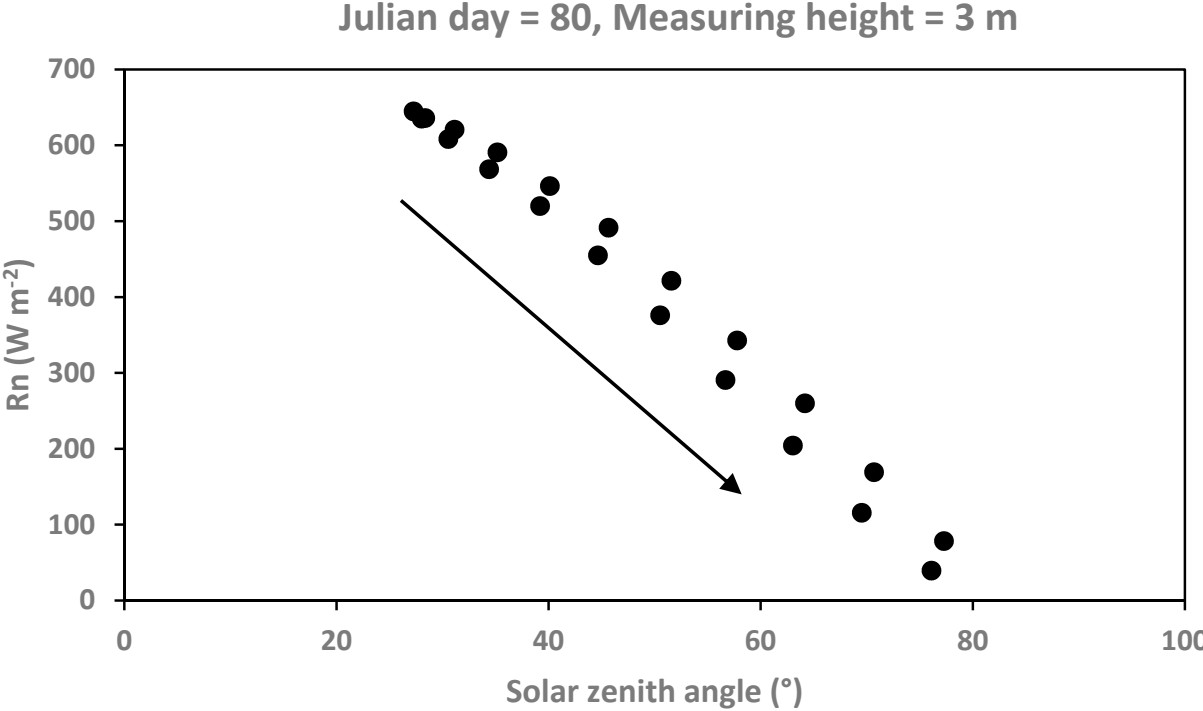

**Figure 12.** Variation of Rn measurements for chickpea (PH4) for Julian days 41 (bare soil) and 80 (maximum fv).

## 4. Discussion

### 4.1. Model Adjustments

The results of the adjustment of the model to measured data show that including or excluding the scale effect had no marked impact on radiative temperature estimates. In contrast, for the albedo estimates including a scale effect improved the experimental fit, as observed with other field measurements [50].

Overall, the OPM model for BRDF (albedo) and BEDF (radiative temperature) adequately fitted ($R^2 > 0.99$) the data from five agricultural crops, measured with different sun-sensor geometry configurations and at different growth stages. This supports the use of the one-parameter model in operations with a single measurement at field or satellite levels.

Using a sun-sensor geometry model (which implies a change to normalized space) can be used to estimate Rn footprint (see below), or to standardize Rn measurements to a fixed sun-sensor geometry, since knowing the parameter g or G, Equation (9) or (10), allows for estimates of any other sun-sensor geometry (angular arguments) analysis variables.

The OPM was developed for minimum data requirements using a transformed (normalized) space. Although the one-parameter model (OPM) can be used with agricultural crops, the problem of using only normalized estimates remains, as using the simple model to produce non-normalized estimates can lead to large errors resulting from the parameterization introduced [49]. An alternative approach (no scale effect) consists of using the footprint model hereby proposed, along with normalized values for all the other components of the energy balance:

$$C_{EB}[Rn \cdot \cos(\chi) - G \cdot \cos(\chi)] = \lambda E \cdot \cos(\chi) + H \cdot \cos(\chi) \tag{18}$$

This equation is exactly the same as in Equation (2). The scale effect implies a logarithmic function for transformations.

### 4.2. Patterns under Sun-Sensor Geometry Variations and Their Implications for the Energy Balance Closure

Radiative temperature measurements decrease at higher zenith angles (as a smaller area of the soil is measured and the foliage contributions increase), whereas albedo measurements increase (as a larger area of vegetation foliage is measured and less soil is contributing). Additionally, the higher the vegetation cover (almost only foliage can be seen), the lower the variation associated with sun-sensor geometry. The OPM model and experimental measurements of Rn (Figure 12) show that albedo variations with sun-sensor geometry are higher than the radiative temperature measurements (Figure 11). Considering that conversion of Tr to Ts can be carried out using a simple adjustment (with atmospheric conditions and fv changing slowly), the surface emissivity had an inverse pattern to that of Tr (Ts) when fv was less than the maximum fv (the pattern change with high fv), but these variations were small [24]. If we use Equation (5), then Rn decreases as the footprint increases (see zenith angle increase) to match the footprints of λE and H. This implies that nadir measurements of Rn are ever overestimated since CEB is less than 1.0, as measured by EC systems [18]. In simple geometric terms, the elliptic footprint of λE and H (normal case) requires a change from a circular to elliptic footprint for Rn (this implies diminishing the value of Rn) to make correct energy and matter balances [17].

If we used the reciprocity principle for BRDF or BEDF [63,64] interchanging solar zenith angles for view ones (Figure 12), then Rn is reduced when view zenith angles are increased (i.e., footprints are larger than nadir view angles as used in EC systems), leading to a mismatch between footprints and a lack of balance between components.

When evaluating the case of term (Rn–G) of the balance defined in Equation (2), different authors, after considering different sources of errors associated with the closure of the energy balance, argue that spatial (and temporal) heterogeneity at the landscape scale could be the cause of a lack of energy and matter [18,19,28]. In order to analyze variations of available energy (Rn–G) it is possible to simplify this term using the linear relation

between Rn and G [65], although this can be more complex [66]. For example, [67] and [68] used the relation G/Rn = c, where c varies with fv. Using this relation, balance of energy of Equation (18) can be modified:

$$C_{EB}[Rn \cdot \cos(\chi) \cdot (1 - c)] = \lambda E \cdot \cos(\chi) + H \cdot \cos(\chi) \tag{19}$$

Although it is possible to estimate the contributions of sun-lit and shaded components variations with a sun-sensor geometry model for G [25], the data requirements for its parameterization are not available using remote sensing. It has been argued that spatial variations at the landscape level are responsible for the no closure. Using Rn measurements, it has been shown that spatial variations with data of homogenous grassland fields are not significant [69], but with complex terrain, these contributions (spatial heterogeneity and the use of Rn measurements at different heights) can explain a major part of the balance of energy closure [21,28,70,71].

Considering that albedo has major contributions to Rn, many authors have argued that with inclined surfaces, it is necessary to correct Rn measurements (horizontally, nadir view) for the slope of the terrain [21,72]. Rn nadir measurements in inclined surfaces are lower than in horizontal ones [73,74] because it is necessary to correct solar radiation for the slope of the surface [72–74]. These corrections show that the conventional Rn measurements in flat terrains are overestimated in complex topography or inclined surfaces. Despite the corrections for inclined surfaces, this approach is only a partial solution to the energy balance closure because it changes the geometry of the terrain and does not consider footprint area adjustments for Rn.

Considering the arguments for the problem of footprint mismatch among components of energy and matter balance (Equation (2)) for any fixed time period, we can set c fixed in Equation (18), so that the lack of closure is simply an issue (geometric version) of comparing measuring area between ellipses and circles. Since Rn always decreases in value as the zenithal angle view increases, this situation ensures overestimating (Rn–G) in the energy balances.

## 5. Conclusions

The dependence of the components of net radiation—radiative temperature and albedo—on sun-sensor geometry was quantified by a simple modeling strategy involving the use of a single observation (additional to angular data) from remote sensors on board satellite platforms. The relationship between sun-sensor geometry and footprints of measurements made with the eddy covariance technique allows for generalization of the sun-sensor geometry model to standardize energy balance footprints and account for the issue of lack-of-closure.

The sun-sensor geometry model of the Rn components presented showed good empirical adjustments with field measurements (albedo (%): $R^2$ = 0.9971, RMSE = 0.432; radiative temperature (°): $R^2$ = 0.9967, RMSE = 0.008). The one-parameter model can be parameterized using only one measurement and sun-sensor geometry data, allowing its operational use.

After analyzing the implications of the developed model and measurements carried out under field conditions, along with footprint geometry, one conclusion associated with the energy balance closure problem is that it can be explained as overestimations due to the nadir view of Rn fluxes along with the mismatch of its footprint with aerodynamic fluxes (latent water and sensible heat fluxes). Improving the precision to estimate the components of the energy balance including sensible heat flux, and consequently the irrigation requirements of crops, will contribute to improving water management.

**Author Contributions:** Conceptualization, Writing—original draft, F.P.; Formal analysis, F.P. and M.I.M.; Data curation, J.G.; Methodology, resources, software, C.W., J.C.R. and J.G.; Writing—review & editing E.A.Y., A.L. and M.A.B.; Investigation M.I.M. and M.A.B. All authors have read and agreed to the published version of the manuscript.

**Funding:** PLEIADES with financial support from the European Commission.

**Institutional Review Board Statement:** Not applicable.

**Informed Consent Statement:** Not applicable.

**Data Availability Statement:** All field data used to validate the sun-sensor geometry are available at http://pmcarbono.org/pmc/bases_datos/BasedatosFootprintAlbedoandTr/.

**Acknowledgments:** This research was conducted as part of the activities of the project Participatory multi-level EO-assisted tools for Irrigation water management and Agriculture Decision-Support (PLEIADES) with financial support from the European Commission.

**Conflicts of Interest:** The funders had no role in the design of the study; in the collection, analyses, or interpretation of data; in the writing of the manuscript, or in the decision to publish the results.

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
