# Peer review of "Angular Modeling of the Components of Net Radiation in Agricultural Crops and Its Implications on Energy Balance Closure"

_water, doi:10.3390/w13213028_

Round 1

Reviewer 1 Report

Manuscript “ANGULAR MODELING OF THE COMPONENTS OF NET RADIATION IN AGRICULTURAL CROPS AND ITS IMPLICATIONS ON ENERGY BALANCE CLOSURE” presented the application of a one-parameter model to  characterize Rn in terms of the albedo and surface temperature/emissivity under a sun-sensor geometry. This is applicable for estimating crop evapotranspiration on large agricultural areas, which in turn is important for water management in irrigated agriculture.

My main question is that: Is the applied parameters in this study are sufficient to solve the problem of water management? We know that there are many parameters that affect the evapotranspiration.

There are some comments and concerns regarding to the submitted manuscript:

  • Keyword: it is not usual to use abbreviations as keywords (OPM, BRDF, BEDF).
  • Page 2 line 90: In this perspective, a …
  • Please add the name of evaluated five crops (Bean, Sorghum, Chickpea, Safflower and Wheat) in the Abstract and Introduction (P3 line 99) sections.
  • Authors reported the accuracy of (R2> 0.99) for their model in five mentioned crops. So, how can you generalize your model to other crops with geometric configuration?
  • Another concern about this study is the study area. As you fitted your model in one region (an agricultural field located in Irrigation District 041-Río Yaqui, in Sonora, Mexico), how can you generalize your model to other regional with different parameters?
  • What about uncertainty of your mode?
  • It is better to add MAPE statistic too (P7 line 239).
  • Conclusion section is not a clear reflection of your findings. I suggest to improve it regarding to your finding and add some suggestions regarding to the water management and solving the problem of water estimation in irrigated agriculture.

Author Response

Response to Reviewer 1 Comments

Point 1: My main question is that: Is the applied parameters in this study are sufficient to solve the problem of water management? We know that there are many parameters that affect the evapotranspiration.

Response 1: The paper provides a simple single parameter model for sun-sensor geometry effects in balance closure using eddy covariance technique. This contributes to improving the precision of measurements, specifically latent heat and in consequence the irrigation demands of crops.

Point 2: Keyword: it is not usual to use abbreviations as keywords (OPM, BRDF, BEDF).

Response 2: BRDF is an abbreviation widely used. OPM and BEDF were replaced with keywords that are not part of the paper name.

Point 3: Please add the name of evaluated five crops (Bean, Sorghum, Chickpea, Safflower and Wheat) in the Abstract and Introduction (P3line 99) sections.

Response 3: The name of evaluated five crops were added in both sections.

Point 4: Authors reported the accuracy of (R> 0.99) for their model in five mentioned crops. So, how can you generalize your model to other crops with geometric configuration?

Response 4: The sun-sensor geometry model is general and it can be applied to any crop and spatial configuration. The g and G parameters change with others crops and configurations. The five crops analysed are contrasting archetypes for crops, to show the validity of the model.

Point 5: Another concern about this study is the study area. As you fitted your model in one region (an agricultural field located in Irrigation District 041-Río Yaqui, in Sonora, Mexico), how can you generalize your model to other regional with different parameters?

Response 5: the answer to this question is the same in Point 4. We have several publications with other crops, grazes and terrestrial ecosystems.

Point 6: What about uncertainty of your model?

Response 6: Different metrics are defined in the paper to show uncertainty of the empirical adjustments of the model. As a general finding, uncertainty was low (see metrics)

Point 7: It is better to add MAPE statistic too (P7 line 239).

Response 7: MAPE and MAE are complementary uncertainty metrics, so adding MAPE show little additional information on uncertainty

Point 8: Conclusion section is not a clear reflection of your findings. I suggest to improve it regarding to your finding and add some suggestions regarding to the water management and solving the problem of water estimation in irrigated agriculture.

Response 8: The proposed information was added.

Reviewer 2 Report

Comments

SUMMARY

The paper addresses the research area related to “Hydrology section” of the MDPI Water journal. I believe that the target journal is an appropriate forum for this article. This paper discusses the issue of energy balance closure resulting from differences in the footprints of the components. The authors proposed a model to characterize Rn in terms of the basic components: albedo and surface temperature/emissivity under a sun-sensor geometry. They also discussed the development of the footprints of these components, based on a simple model parameterized under sun-sensor geometry considerations.

BROAD COMMENT

The Introduction section is well written with recent references. I appreciate the fact that the authors described in detail the methodology used in the study. It helps to grasp the message they are sharing in the paper. They also discussed well the results of the study. However, the authors failed to use at least two years of experiment data to capture the variabilities and the errors which may arise from the experimental set-up, please mention this in the limitations of the study. The authors failed to put the implications of the results of the study in a big picture to show the contribution of the study to research.

SPECIFIC COMMENTS

  • Lines 239-240: The equation 12 is two equations, so I suggest the authors number them 12 and 13.
  • Lines 355-356: Please, put the units of the data on both axes in their captions.
  • Lines 370-371: Please, label the graphs on Figure 8, (a), (b), ..etc
  • Lines 384-385: Please, label the graphs on Figure 9, (a), (b) and put the units of the data on both axes in their captions.
  • Lines 386-387: Please, label the graphs on Figure 10, (a), (b) and put the units of the data on both axes in their captions.
  • Lines 397-398: Please, label the graphs on Figure 11, (a), (b) and put the units of the data on both axes in their captions.

Author Response

Response to Reviewer 2 Comments

BROAD COMMENTS

Point 1: the authors failed to use at least two years of experiment data to capture the variabilities and the errors which may arise from the experimental set-up, please mention this in the limitations of the study.

Response 1: Only data for an agricultural cycle was considered, with the monitoring of five different and contrasting crops. It is not necessary to use several years of experimental data to validate the introduced model. Also, it is not an experimental approach to showing variations over g and G parameters of the model, but only to demonstrate that defining energy balance closure can be formatted in terms of sun-sensor geometry to analyse differences in closure and to showing that imbalance is a simply consequence of using different footprints.

Point 2: The authors failed to put the implications of the results of the study in a big picture to show the contribution of the study to research.

Response 2: The conclusions were expanded by adding the implications of the study in relation to water management and energy closure.

SPECIFIC COMMENTS

Point 3: Lines 239-240: The equation 12 is two equations, so I suggest the authors number them 12 and 13.

Response 3: they were separated into two equations: 12 and 13.

Point 4: Lines 355-356: Please, put the units of the data on both axes in their captions.

Response 4: We write the units in Figure 6.

Point 5: Lines 370-371: Please, label the graphs on Figure 8, (a), (b), etc.

Response 5: The labels are in the top of each graph. The Figure 8 shows the variations in radiative temperature (were up to 10 °C) and the albedo values (variations between 10 and 40%) at the same condition of crops, that is to say, the same above-ground coverage (fv) and Julian day.

Point 6: What about uncertainty of your model?

Response 6: we present different uncertainty metrics in empirical adjustments of the model.

Point 7: Lines 384-385: Please, label the graphs on Figure 9, (a), (b) and put the units of the data on both axes in their captions.

Response 7: We write the units in Figure 9.

Point 8: Lines 386-387: Please, label the graphs on Figure 10, (a), (b) and put the units of the data on both axes in their captions.

Response 8: We write the units in Figure 10.

Point 9: Lines 397-398: Please, label the graphs on Figure 11, (a), (b) and put the units of the data on both axes in their captions.

Response 9: We write the units in Figure 11 and the labels are in the top of each graph.

Round 2

Reviewer 1 Report

All comments have been done.

Author Response

Response to Reviewer 1 Comments

Point 1: My main question is that: Is the applied parameters in this study are sufficient to solve the problem of water management? We know that there are many parameters that affect the evapotranspiration.

Response 1: The paper provides a simple single parameter model for sun-sensor geometry effects in balance closure using eddy covariance technique. This contributes to improving the precision of measurements, specifically latent heat and in consequence the irrigation demands of crops.

Point 2: Keyword: it is not usual to use abbreviations as keywords (OPM, BRDF, BEDF).

Response 2: BRDF, OPM and BEDF were replaced with keywords that are not part of the paper name.

Point 3: Please add the name of evaluated five crops (Bean, Sorghum, Chickpea, Safflower and Wheat) in the Abstract and Introduction (P3line 99) sections.

Response 3: The name of evaluated five crops were added in both sections.

Point 4: Authors reported the accuracy of (R> 0.99) for their model in five mentioned crops. So, how can you generalize your model to other crops with geometric configuration?

Response 4: The sun-sensor geometry model is general and it can be applied to any crop and spatial configuration. The g and G parameters change with others crops and configurations. The five crops analysed are contrasting archetypes for crops, to show the validity of the model.

In page 7, lines 235-238, we mention “The OPM has been validated for reflectance measurements in experimental settings in laboratory [51,52], and with variations in the angular vision of satellite reflectance [53,54], while also being used to estimate more complex BRDF models [55]; with good results throughout (generally, R2 > 0.99)”

Point 5: Another concern about this study is the study area. As you fitted your model in one region (an agricultural field located in Irrigation District 041-Río Yaqui, in Sonora, Mexico), how can you generalize your model to other regional with different parameters?

Response 5: the answer to this question is the same in Point 4. We have several publications with other crops, grazes and terrestrial ecosystems (see note about references mentioned in text)

Point 6: What about uncertainty of your model?

Response 6: Different metrics are defined in the paper to show uncertainty of the empirical adjustments of the model. As a general finding, uncertainty (definition depend on metrics and statistical models used) was low (see metrics R2, RMSE, EAM, ERM)

Point 7: It is better to add MAPE statistic too (P7 line 239).

Response 7: MAPE (MRE) and MAE are complementary uncertainty metrics, so adding MAPE show little additional information on uncertainty. We include MRE metrics

Point 8: Conclusion section is not a clear reflection of your findings. I suggest to improve it regarding to your finding and add some suggestions regarding to the water management and solving the problem of water estimation in irrigated agriculture.

Response 8: The proposed information was added.
